# Physical Exercise, Social Capital, Hope, and Subjective Well-Being in China: A Parallel Mediation Analysis

**DOI:** 10.3390/ijerph20010303

**Published:** 2022-12-25

**Authors:** Xupeng Zhang, Dianxi Wang, Fei Li

**Affiliations:** School of Marxism, Beijing Sport University, Beijing 100084, China

**Keywords:** subjective well-being, physical exercise, social network, sense of hope, sport

## Abstract

Based on data from the 2017 China General Social Survey, a conditional process analysis was conducted to explore the association between physical exercise and subjective well-being, as well as the parallel mediating effect of social networks and a sense of hope. The results showed that physical exercise had a significant positive predictive effect on the participants’ subjective well-being. Furthermore, social networks and a sense of hope mediated the association between physical exercise and subjective well-being. Physical exercise indirectly promoted subjective well-being by enhancing social networks and a sense of hope. However, the indirect effect of a sense of hope on the association between physical exercise and subjective well-being was greater than that of social networks. The results of this study revealed the internal mechanism of the effect of participation in physical exercise on the improvement of subjective well-being, which is of great significance for formulating relevant policies and plans to further improve Chinese residents’ well-being.

## 1. Introduction

Subjective well-being refers to an individual’s affirmative attitude and feeling that their current living condition aligns with their ideal life [1]. Thus, governments and international organizations aim to improve residents’ well-being through various actions and plans, such as reducing poverty and inequality, improving infrastructure and the environment, strengthening public-health services, and increasing their capacity [2,3,4,5,6]. Since the economic reform and opening-up of China in 1978, which has contributed to sustained and rapid economic growth, the living standards of residents have significantly improved. For instance, the per capita disposable income of Chinese households has increased from CNY 171 in 1978 to CNY 32,189 in 2020 [7]. Meanwhile, residents’ sense of happiness in life has also been continually reinforced via the continuous improvement in their standard of living. According to the World Happiness Report released by the United Nations, among 156 countries and regions in the world, China has improved its rank from the 112th position in 2012 to the 72nd in 2022 [8,9]. 

Previous studies showed that exercise effectively maintains and promotes subjective well-being. For example, using the revised version of the Brief Subjective Well-Being Scale for Chinese Citizens, scholars found that the exercise duration and frequency of members of sports associations positively predict subjective well-being [10]. Kim et al. [11] found that sports participation is significantly and positively associated with general feelings of happiness. In addition, Downward et al. [12] found that sports participation, mediated through health, can improve overall subjective well-being. Thus, exercise can improve and maintain residents’ subjective well-being. However, although studies have demonstrated a positive association between exercise and subjective well-being, the path through which participation in physical exercise improves subjective well-being is yet to be clarified. To address this gap in the literature, this study used data from the 2017 China General Social Survey (CGSS) to examine social networks and a sense of hope as mediating variables in the association between participation in physical exercise and subjective well-being. We assumed that residents’ participation in physical exercise would expand their social networks and enhance their sense of hope, thereby affecting their subjective well-being. Therefore, this study aimed to reveal how participation in physical exercise affects subjective well-being.

## 2. Theoretical Background

### 2.1. Physical Exercise and Subjective Well-Being

In the extant literature, scholars have found that participation in physical exercise is associated with personal health, subjective well-being, and quality of life [13,14,15,16,17,18]. Participating in physical exercise not only improves residents’ subjective well-being but also has a significant positive effect on their physical and psychological health [14]. Furthermore, the duration and frequency of exercise also have a positive predictive effect on subjective well-being [10,19]. 

Subjective well-being comprises positive and negative emotional experiences and life satisfaction at the cognitive level [20]. Additionally, regular participation in physical exercise affects life satisfaction. For example, some scholars have found a highly significant positive correlation between the intensity of physical activity, cognition of the importance of physical activity, and various dimensions of life satisfaction [21,22]. In addition, when the frequency of physical exercise is maintained at 1–3 times a month, life satisfaction begins to increase; when the frequency of physical exercise is maintained at 3–4 times a week, satisfaction reaches its highest point and begins to decline as the frequency of exercise increases [23]. Therefore, following previous research, we assumed that individual subjective well-being is positively affected by physical activity. Consequently, we proposed research hypothesis 1. 

**Research Hypothesis 1 (H1):** *Residents’ physical exercise and subjective well-being are positively correlated*.

### 2.2. Social Capital and Subjective Well-Being

Physical exercise can have social attributes; participating in it not only improves physical fitness but can also cultivate social networks [24,25,26]. Based on the social nature of sports, sports are often regarded as positive forms of social capital conducive to solving social problems and increasing social cohesion, trust, and connection. Establishing social capital through sports associations has become a common way of enhancing social solidarity, generating mutually beneficial connections, and building trust in others, further increasing social cohesion and eliminating social exclusion. For instance, community sports participation can expand social networks, and those residents who regularly participate in physical exercise have social networks with high levels of interaction and intimacy [25]. Therefore, physical exercise improves physical and mental health and emotional experiences at the individual level and can expand interpersonal communication and relationship networks, thereby encouraging social solidarity.

A growing body of research has investigated the association between social capital and well-being. The relatively consistent findings of these studies show that social capital positively affects people’s subjective well-being and quality of life; that is, the richer the social capital, the higher people’s subjective well-being and quality of life [27,28,29,30,31]. Furthermore, the significant effects of social networks and social interaction on subjective well-being have been verified for different compositions of social capital. Studies have found that social-network size positively affects residents’ subjective well-being [32,33], and people with embedded network types characterized by greater social capital tend to report greater well-being [34]. In China, Ma [35] found that the proportion of friends in Chinese New Year greetings, the frequency of neighborhood interactions, and informal social participation positively affected life satisfaction.

Studies have also found that social networks may mediate the relationship between participation in sports and subjective well-being [11], and social interaction with others can lead to well-being during physical exercise [36]. Furthermore, neighborhood ties were found to positively mediate the relationship between sports participation and overall well-being [11], and peer interaction partially mediated the link between physical exercise and subjective well-being [37]. According to this analysis, social capital has an important effect on subjective well-being, while the impact of physical exercise on subjective well-being varies with the level of social capital. Consequently, social capital may mediate the association between participation in physical exercise and subjective well-being. Therefore, we proposed the second hypothesis. 

**Research Hypothesis 2 (H2):** *Social networks mediate the association between physical exercise and subjective well-being*.

### 2.3. Sense of Hope and Subjective Well-Being

Hope refers to a person’s capacity to identify pathways to goals and achieve the desired outcomes of goal-oriented pursuits [38,39]. Previous studies found that hope is associated with subjective well-being; a high level of hope can positively affect well-being [40,41,42,43,44]. Hope theory is often used to explain the link between hope and subjective well-being, which posits that hope results in higher levels of subjective well-being when individuals persist in pursuing their goals. Those who are more hopeful show greater perseverance in pursuing their targets, which can lead to higher levels of happiness through more successful experiences [38,41].

Furthermore, Pleeging et al. [45] pointed out that behavior and perception explain the relationship between hope and subjective well-being. People with higher hopes can foresee more opportunities and tend to invest in their future, thereby exhibiting positive behaviors to achieve things that make them happy with their lives and result in favorable outcomes. For example, studies have shown that a higher level of hope predicts outstanding academic and professional achievement [46,47], higher self-esteem and optimism [48], and better psychological and physical health outcomes [43,49]. Moreover, people with high levels of hope are more inclined to incorporate positive emotions into their subjective well-being and perceive hope in their daily lives. People who maintain positive emotions are more open to new circumstances, relationships, and impressions and can obtain more experience and skills [45,50]. Additionally, hope may be a protective factor against adverse outcomes, thereby reducing the negative consequences of frustration [51], alleviating anxiety and depressive symptoms [52], and buffering the adverse impact of the COVID-19 pandemic on well-being [53].

Hope is also often viewed as a link between specific factors influencing subjective well-being. Studies have identified the mediating effect of hope on the relationship between subjective well-being and specific factors, such as income [45], attachment [54], perceived power [29], sense of coherence [55,56], positive emotions [57], and psychological vulnerability [58,59,60]. As an important dimension of psychological capital, hope may also play a crucial part in the association between physical exercise and subjective well-being. For example, one study found that hope was vital in the relationship between injured athletes’ recovery beliefs and subjective well-being [61]. Another study found that psychological capital partially mediated the relationship between exercise duration and subjective well-being [10]. Given the findings of previous studies, we assumed that hope would mediate the association between physical exercise and subjective well-being. Therefore, we proposed a third hypothesis. 

**Research Hypothesis 3 (H3):** *Sense of hope mediates the association between physical exercise and subjective well-being*.

### 2.4. The Current Study

Various studies have conducted extensive analyses on the association between physical exercise and subjective well-being. However, few studies have examined the associations between physical exercise, social networks, a sense of hope, and subjective well-being based on Chinese data. Therefore, based on the literature on the associations between physical exercise, social networks, a sense of hope, and subjective well-being, we formed the theoretical framework of this study, as shown in Figure 1. Specifically, we hypothesized that participation in physical exercise would not only directly enhance residents’ subjective well-being but also indirectly affect subjective well-being by encouraging the extroverted acquisition of social capital and a sense of hope.

## 3. Methods

### 3.1. Data

Data from the 2017 CGSS were applied in this study. The CGSS is a large-scale, comprehensive social-survey project sponsored by the National Survey Research Center at the Renmin University of China. Since 2003, the CGSS has conducted sample surveys of more than 10,000 households across the country every year. It is the earliest national, comprehensive, and continuous academic survey project in China. The 2017 CGSS adopted multi-stage stratified sampling. The 12,852 participants interviewed were from 29 provinces, municipalities, and autonomous regions across China. Our study was approved by the institutional review board of the Renmin University of China. Moreover, we obtained written informed consent from the participants. The Well-Being Scale was only administered to specific participants, resulting in missing data in the sample. Therefore, cases with missing data were deleted from this study, and only questionnaires with complete information were retained. Consequently, the final sample comprised 4031 participants. The subjective well-being measurement items belong to the fourth survey module of the CGSS 2017, and this module is specially set up for the horizontal international comparative study with the East Asian Social Survey 2016. This module interviewed 6000 people, 4132 of whom had valid cases. After excluding the cases that did not respond to the happiness-scale survey and the missing values of other research variables in our study, the final sample was 4031. Table 1 shows the demographic characteristics of the final sample.

### 3.2. Variables

Dependent variable. The dependent variable in this study was subjective well-being. The CGSS uses the Brief Subjective Well-Being Scale for Chinese Citizens, created by Xing [62], to measure subjective well-being. The scale comprises 20 items across 10 dimensions: contentment and abundance, mental health, social confidence, growth and progress, target values, self-acceptance, physical health, mental balance, interpersonal adaptation, and family atmosphere. The items are scored on a 6-point scale, ranging from 1 (“strongly disagree”) to 6 (“strongly agree”). After converting the responses to items that were reverse-scored, subjective well-being scores were obtained by adding the values of each item. The maximum and minimum values were 120 and 25, respectively. The scale’s Cronbach’s α was 0.849.

Independent variable. Physical exercise was the independent variable in this study and was measured with the item asking, “In the past 12 months, how many times per week did you normally perform up to 30 min of physical activity that made you sweat?” This is a continuous variable with a maximum value of 96, a minimum value of 0, and an average of 2.335.

Mediating variables. Participants’ social network was set as a mediating variable. Following Li and Chen [63], three questions were selected to assess participants’ social networks: “How often do you engage in social entertainment activities with your neighbors?” “How often do you have social entertainment activities with friends?” and “How often do you get together with relatives who do not live together?” Answers to these questions ranged from 1 (“never”) to 7 (“almost every day”). The social-network score was obtained by adding the values of the three indicators. This is a continuous variable with a maximum value of 19, a minimum value of 3, and an average value of 10.081.

A sense of hope was also set as a mediating variable. The State Hope Scale, developed by Snyder et al. [64], measured participants’ sense of hope. This scale consists of six items: “If I find myself in a jam, I can think of many ways to get out of it”, “Currently, I am energetically pursuing my goals”, “There are several ways around any problem that I may face”, “Right now, I see myself as being pretty successful”, “I can think of many ways to reach my current goals”, and “Currently, I am meeting the goals that I have set for myself.” The responses to these items ranged from 1 (“definitely false”) to 8 (“definitely true”). The hope score was obtained by summing the values of these items.

Control variables. Previous studies found significant individual differences in subjective well-being according to gender, age, race, and socioeconomic status [19,65,66,67,68]. In the Chinese context, hukou (a categorization based on one’s household registration) and Communist Party membership are important dimensions to consider, both as important indicators of social stratification and socioeconomic status in China [69,70] and as factors that may affect the subjective well-being of Chinese people. For example, a prior study found an obvious difference between urban and rural hukou in the subjective well-being of Chinese residents [71]. In addition, as a prestigious social identity, Chinese Communist Party membership significantly influenced subjective well-being [72]. Therefore, gender, age, ethnicity, hukou status, education level, marital status, political status, and occupation were included in the analysis as control variables. Gender was a categorical variable, coded as 1 = male, 2 = female. Age was a continuous variable with 18, 96, and 50.754 as the minimum, the maximum, and the average age. Ethnicity was a categorical variable, with 1 indicating an ethnic minority and 2 indicating Han ethnicity. Hukou were divided into “rural hukou” (1) and “urban hukou” (2), which were in turn divided into four categories: “elementary and below” (1), “middle school” (2), “high school” (3), and “college or above” (4). Marital status included “no spouse” (1) and “married” (2). Political status included “non-party member” (1) and “party member” (2).

### 3.3. Statistical Methods

A descriptive analysis was performed to provide a basic overview of the sample. Independent sample *t*-tests and F-tests were performed to compare the mean differences of demographic variables for the key study variables. In this study, conditional process analysis (CPA), proposed by Hayes [73], was used to examine the mediating effects of social capital and a sense of hope. Specifically, Model 4 (i.e., the simple mediation model) in the SPSS macro compiled by Hayes [74] was used to estimate the mediating effects of social networks and a sense of hope on the association between physical exercise and subjective well-being. The bootstrapping method was used to estimate 95% confidence intervals based on 5000 random samples to test the significance of the mediating effect. When the 95% confidence interval did not contain 0, the result was considered statistically significant [75]. The model equation for fit is as follows:(1)$$Y=\lambda_{0}+\lambda_{1}X+\lambda_{2}C_{1}$$
(2)$$M=\pi_{0}+\pi_{1}X+\pi_{2}C_{2}$$
(3)$$Y=\beta_{0}+\beta_{1}X+\beta_{2}M+\beta_{3}C_{2}$$
where *X*, *M*, and *Y* represent independent, mediating, and dependent variables, respectively; *C*_1_, *C*_2_, and *C*_3_ are control variables; $\lambda_{1}$ represents the total effect of the independent variable on the dependent variable; and $\beta_{1}$ represents the direct effect of the independent variable on the dependent variable.

## 4. Results

### 4.1. Descriptive Statistical Analysis

Table 2 presents the differences in participants’ subjective well-being according to their demographic and social characteristics. Men’s subjective well-being was significantly higher than that of women. Subjective well-being decreased with age; the subjective well-being of the older adults was lower than that of the younger participants. Although the subjective well-being of the Han participants was higher than that of the minority participants, the difference was not statistically significant. The subjective well-being of the urban hukou residents was significantly higher than that of the rural hukou residents. The subjective well-being of the participants with party membership was significantly higher than that of those without party membership. Subjective well-being increased on par with educational attainment, and the married participants’ subjective well-being was significantly higher than that of unmarried participants. The results of the descriptive statistical analysis showed that participants’ subjective well-being varied by gender, age, hukou, political status, education level, and marital status.

Table 3 shows the Pearson correlation coefficients between the key variables. Physical exercise was positively associated with social network, a sense of hope, and subjective well-being. The social network was also positively correlated with a sense of hope and subjective well-being; however, the correlation coefficient was small. A sense of hope was positively related to subjective well-being. As the Pearson correlation only reflects the bivariate correlation between variables, a CPA analysis was needed to provide stronger support for hypothesis testing.

### 4.2. Conditional Process Analysis

In this study, Process V3.3, an SPSS plug-in compiled by Hayes [74], was used. The mediating effect of social networks on the association between physical exercise and subjective well-being was tested while controlling for gender, age, ethnicity, hukou status, political status, education level, and marital status, as shown in Table 4 and Table 5. Based on Model 1, physical exercise had a significant positive impact on subjective well-being; at a significance level of 0.001, the participants’ subjective well-being score increased by 0.206 for every unit of physical exercise. In Model 2, the positive effect of physical exercise on social networks was significant, and the social-network score increased by 0.057 for every unit of physical exercise. In Model 3, physical exercise had a significant positive effect on the sense of hope, and the sense-of-hope score increased by 0.182 for each unit increase in physical exercise. According to Model 4, when the mediating variables of social network and sense of hope were included, the direct effect of physical exercise on subjective well-being was marginally significant; that is, after controlling for the mediating variables and demographic social variables, the respondents’ subjective well-being score increased by 0.065 for every unit of physical exercise. In addition, in Model 4, both social networks and a sense of hope had significant positive effects on subjective well-being.

Table 5 presents the results of the mediation-effect tests. The upper and lower limits of the bootstrap 95% confidence interval for the direct effect of physical exercise on subjective well-being and the mediating effect of social network and sense of hope do not contain 0, indicating that physical exercise not only directly affected subjective well-being but also affected subjective well-being through the mediating effect of social network and sense of hope. The direct effect of physical exercise on subjective well-being was 0.064, accounting for 31.07% of the total effect. Furthermore, the mediating effect of the social network was 0.011, and that of the sense of hope was 0.131, accounting for 5.34% and 63.59% of the total effect, respectively. Additionally, the sum of the mediation effects was 0.142, forming 68.93% of the total effect.

## 5. Discussion

This study investigated the mediating effect of social networks and a sense of hope on the association between physical exercise and subjective well-being based on data from the 2017 CGSS. First, the results show that physical exercise had a significant positive effect on subjective well-being after controlling for gender, age, ethnicity, hukou status, political status, education level, and marital status. Second, social networks and a sense of hope mediated the association between physical exercise and subjective well-being. Physical exercise may enhance individuals’ subjective well-being by expanding their social networks and improving their sense of hope. Third, the mediating effect of a sense of hope on the association between physical exercise and subjective well-being was greater than that of social networks.

The findings of this study are consistent with previous studies, which found that participation in physical exercise predicts subjective well-being [14,15,17,18]. The mediating role of social networks in the association between physical exercise and subjective well-being is consistent with the findings of Liu [76] and Kim et al. [11]. Therefore, physical exercise directly and positively affects residents’ subjective well-being through their social networks. In addition, this study found that the participants’ sense of hope had a mediating role between physical exercise and subjective well-being; thus, participating in physical exercise is beneficial for cultivating positive emotions and promoting subjective well-being. This finding lends support to the idea that positive emotions play a mediating role in the association between physical exercise and subjective well-being [19,61]. The main contribution of this study is that it further clarifies the mechanism between physical-exercise participation and subjective well-being while also revealing the role of social networks and a sense of hope in this association, thereby enriching research on the association between physical exercise and subjective well-being.

Based on these findings, the possible explanations for how physical exercise increases subjective well-being through social networks and a sense of hope are as follows. By participating in physical exercise, individuals develop trusting relationships and expand their social capital, which is conducive to enhancing their subjective well-being. Nevertheless, people may also cultivate and form positive emotions to maintain a healthy and optimistic state of mind, which is conducive to improving subjective well-being. In other words, sports and exercise are efficient platforms for individuals to expand their social networks and constitute an important means of promoting happiness.

This study has certain limitations. First, because the 2017 CGSS did not have data on the subjective well-being of some of its participants, we were unable to secure data on all the participants who participated in the national survey. Consequently, the validity of the findings may be limited. Second, the use of cross-sectional data to explain the association between physical exercise and subjective well-being is another limitation of this study. Long-term longitudinal data may reveal a more accurate causal relationship. Third, the mediating effects of social networks and the sense of hope on physical-exercise participation and subjective well-being may vary according to participants’ gender, age, place of residence (urban vs. rural), and education level. Future research could further explore the heterogeneity of the mediating mechanism of social networks and sense of hope for physical exercise and subjective well-being among groups with greater diversity in terms of gender, place of residence, age, and education level.

## 6. Conclusions

Using CPA, this study found that physical exercise has both direct and indirect effects on residents’ subjective well-being. Physical exercise not only directly and positively affects residents’ subjective well-being but also indirectly affects residents’ subjective well-being through social networks and a sense of hope as mediating variables. Although an increase in physical exercise may lead to an improvement in subjective well-being, increasing physical exercise may also indirectly improve subjective well-being by enhancing the social network and sense of hope. In other words, as mediators of physical exercise and subjective well-being, social networks and a sense of hope may enhance the positive effect of physical exercise on subjective well-being by strengthening the link between physical exercise, social networks, and a sense of hope.

This study has several policy implications. First, the government should strengthen the construction of physical-exercise facilities in grassroots communities. Based on the actual fitness needs of community residents, the construction and transformation of community physical-exercise facilities should be strengthened to provide infrastructure conditions for all kinds of people to perform physical exercise in the community and constantly improve the accessibility and convenience of community physical exercise facilities. Second, the government must improve the network of grassroots sports organizations, promote the diversification of sports community organizations, cultivate community communication networks, enrich social capital, and provide organizational support for community sports participation and happiness promotion. Third, the government, society, and families should cultivate and enhance residents’ psychological capital, strengthen psychological support, thereby improving subjective well-being.

## Figures and Tables

**Figure 1 ijerph-20-00303-f001:**
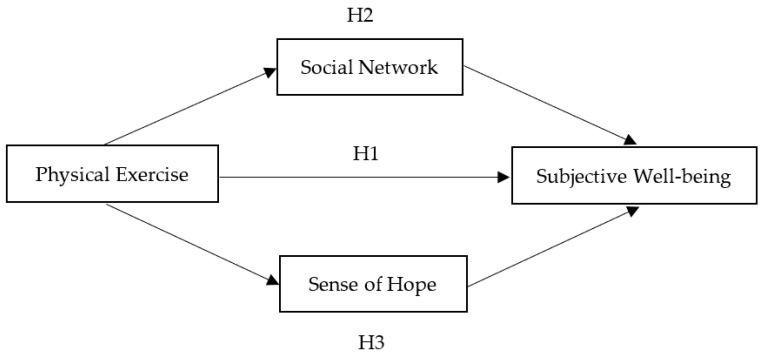
The parallel-mediation model.

**Table 1 ijerph-20-00303-t001:** Sample characteristics.

		Original Sample (N = 12,852)		Subsample of SWB Survey (N = 4132)		Final Valid Sample of This Study (N = 4031)	
Variables		Frequency	Percent (%)	Frequency	Percent (%)	Frequency	Percent (%)
Sex	Male	5935	47.17	1916	46.37	1872	46.44
	Female	6647	52.83	2216	53.63	2159	53.56
Age group							
	under 35 years	2766	21.98	909	22.00	892	22.13
	35–45 years	1942	15.43	623	15.08	612	15.18
	46–60 years	3821	30.37	1258	30.45	1238	30.71
	over 60 years	4053	32.21	1342	32.48	1289	31.98
Ethnic	Minority ethnic groups	946	7.52	302	7.31	298	7.39
	Han ethnic groups	11,636	92.48	3830	92.69	3733	92.61
Hukou type	Rural	6777	53.86	2170	52.52	2109	52.32
	Urban	5805	46.14	1962	47.48	1922	47.68
Education level	Elementary and below	4345	34.53	1393	33.71	1336	33.14
	Middle school	3511	27.90	1160	28.07	1139	28.26
	High school	2252	17.90	751	18.18	740	18.36
	College or above	2470	19.63	828	20.04	816	20.24
	Missing values	4	0.03	—	—	—	—
Marital status	No spouse	2934	23.32	987	23.89	951	23.59
	Have a spouse	9648	76.68	3145	76.11	3080	76.41
Political status	Non-party member	10,512	83.55	3462	83.79	3373	83.68
	Party member	2060	16.37	669	16.19	658	16.32
	Missing values	10	0.08	1	0.02	—	—

**Table 2 ijerph-20-00303-t002:** Differences in subjective well-being of respondents with different characteristics (N = 4031).

	Mean ± SD	Test	*p*
Sex:			
Male	82.622 ± 12.502	*T* = 2.440	*p* = 0.015
Female	81.654 ± 12.611		
Age group:			
under 35 years	84.860 ± 11.246	*F* = 28.56	*p* < 0.001
35–45 years	83.734 ± 11.176		
45–60 years	80.972 ± 12.904		
over 60 years	80.510 ± 13.315		
Ethnic:			
Minority ethnic groups	81.027 ± 12.786	*T* = −1.538	*p* = 0.124
Han ethnic groups	82.190 ± 12.549		
Hukou type:			
Rural	79.800 ± 12.165	*T* = −12.419	*p* < 0.001
Urban	84.632 ± 12.521		
Political status:			
Non-party member	81.131 ± 12.468	*T* = −11.301	*p* < 0.001
Party member	87.091 ± 11.887		
Education level:			
Elementary and below	77.353 ± 12.721	*F* = 126.46	*p* < 0.001
Middle school	82.597 ± 12.021		
High school	84.396 ± 11.554		
College or above	87.115 ± 11.223		
Marital status:			
No spouse	80.739 ± 13.841	*T* = −3.837	*p* < 0.001
Have a spouse	82.525 ± 12.120		

**Table 3 ijerph-20-00303-t003:** Pearson correlation analysis.

		1	2	3	4
1	Physical exercise	1			
2	Social network	0.063 ***	1		
3	Sense of hope	0.130 ***	0.081 ***	1	
4	Subjective well-being	0.121 ***	0.085 ***	0.497 ***	1

Note: *** *p* < 0.001.

**Table 4 ijerph-20-00303-t004:** Results of mediating-effect model (N = 4031).

	Model 1: Subjective Well-Being		Model 2: Social Network		Model 3: Sense of Hope		Model 4: Subjective Well-Being	
	B	SE	B	SE	B	SE	B	SE
Physical exercise:	0.206 ***	0.040	0.057 ***	0.012	0.182 ***	0.024	0.065 ^†^	0.036
Social network:							0.190 ***	0.047
Sense of hope:							0.718 ***	0.023
Sex: (Ref.: Male)	−0.051	0.380	0.294 **	0.113	−0.435 ^†^	0.234	0.206	0.340
Age:	−0.022	0.013	−0.017 ***	0.004	−0.097 ***	0.008	0.051 ***	0.012
Hukou type: (Ref.: Rural)	1.761 ***	0.452	−0.552 ***	0.135	0.366	0.278	1.603 ***	0.405
Education level: (Ref.: Elementary and below)								
Middle school	4.172 ***	0.518	0.311 *	0.154	1.745 ***	0.318	2.861 ***	0.465
High school	5.136 ***	0.634	0.261	0.189	1.916 ***	0.390	3.712 ***	0.569
College or above	7.087 ***	0.736	−0.235	0.219	3.955 ***	0.453	4.294 ***	0.665
Marital status: (Ref.: No spouse)	2.729 ***	0.449	0.085	0.134	1.812 ***	0.276	1.413 ***	0.404
Ethnic: (Ref.: Minority ethnic groups)	0.006	0.719	0.019	0.214	−0.963 *	0.442	0.694	0.643
Political status: (Ref.: Non-party member)	3.214 ***	0.552	0.209	0.164	0.892 **	0.339	2.534 ***	0.494
Constant	68.087 ***	1.990	10.633 ***	0.593	27.075 ***	1.223	46.644 ***	1.941
R-sq	0.114		0.021		0.015		0.290	
F	50.602 ***		8.602 ***		67.943 ***		136.976 ***	
DF	10		10		10		12	

Note: * *p* < 0.05; ** *p* < 0.01; *** *p* < 0.001; ^†^ < 0.1.

**Table 5 ijerph-20-00303-t005:** Total effect, direct effect and mediating effect.

	Effect	Boot SE	Boot CI Lower	Boot CI Upper	Relative Effect Ratio
Total effect	0.206	0.051	0.138	0.339	
Direct effect	0.064	0.038	0.003	0.154	31.07%
Mediating effect of social network	0.011	0.004	0.004	0.021	5.34%
Mediating effect of sense of hope	0.131	0.022	0.094	0.185	63.59%

## Data Availability

The datasets used and/or analyzed during the current study are available from the official website of CGSS, http://www.cnsda.org/index.php?r=projects/view&id=94525591 (accessed on 10 July 2022).

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
