# Peer review of "Physical Exercise, Social Capital, Hope, and Subjective Well-Being in China: A Parallel Mediation Analysis"

_ijerph, 2022, doi:10.3390/ijerph20010303_

Round 1
Reviewer 1 Report
Dear authors, I congratulate you on the research carried out since it provides empirical evidence to the field. It is an important advance for society and the scientific community.
The article presents research questions, which are addressed throughout the entire text.
To explore the relationship between physical exercise and subjective well-being, as 9 well as the parallel mediating effect of social networks and sense of hope
I consider the topic original or relevant in the field, and I consider that the article addresses a specific gap in the field of study.
The article provides evidence on physical exercise, social capital, hope and subjective well-being in China: a parallel mediation analysis. This evidence provides important elements for the implementation of programs in the field of mental health and the promotion of well-being.
The document is clear on a methodological level. However, it is important to review the issue of lost data and the limitations given in the article because this would make it possible to demonstrate the strategy used to select data without bias.
The references are appropriate for the study. These are in line with what was proposed in the study.
Reviewer 2 Report
The paper relates an interesting approach regarding Chinese people health and can be improved to increase its scientific value.
Some remarks regarding the content:
- The abstract and the conclusions must be rephrased and must be included some obtained results of the manuscript, matching the paper hypothesis. In this version of the manuscript no scientific arguments are related.
- Some sentences must be rephrased enterally due to their lack of understanding. Some examples are at lines 34, 43, 47. Also the English language must be revised for the entire manuscript.
- How the collected data express the connection between the 3 chosen hypothesis? Is there any connection between those 3 hypothesis? Can be used other hypothesis in this case beside those presented in the manuscript?
Reviewer 3 Report
The article is devoted to an urgent topic – the study of the state of social well-being and physical exercise in a society that has quickly reached a high standard of living on the example of China. In general, the work is relevant and of interest to readers and corresponds to the subject of the journal. The methodological methods of research used by the authors are adequate to the tasks of the work.
At the same time, there are a number of comments that, in the opinion of the reviewer, require correction. Although the reviewer is not a native English speaker, after reading the work, the impression was created about the need for stylistic and editorial editing of the text. Unfortunately, in some cases, the authors use not quite scientific turns of speech – (L34 – a large body (??) of literature), P110 –“hope references". The term "hukoy type" was used several times in the text – I think not all readers will immediately determine what it is about.
Thus, the work requires mandatory editorial editing and refinement of the text itself in terms of terminology refinement
